# Learning to Filter Outlier Edges in Global Structure-from-Motion

## Abstract

This paper introduces a novel approach to improve camera position estimation in global Structure-from-Motion (SfM) frameworks by filtering inaccurate pose graph edges, representing relative translation estimates, before applying translation averaging. In SfM, pose graph vertices represent cameras and edges relative poses (rotation and translation) between cameras. We formulate the edge filtering problem as a vertex filtering in the dual graph – a line graph where the vertices stem from edges in the original graph and the edges from cameras. Exploiting such a representation, we frame the problem as a binary classification over nodes in the dual graph. To learn such a classification and find outlier edges, we employ a Transformer architecture-based technique. To address the challenge of memory overflow often caused by converting to a line graph, we introduce a clustering-based graph processing approach, enabling the application of our method to arbitrarily large pose graphs. The proposed method outperforms existing relative translation filtering techniques in terms of final camera position accuracy and can be seamlessly integrated with any other filters. The source code will be made public.

## 1 Introduction

The task of generating 3D models from large sets of unordered images poses a significant challenge within computer vision and robotics, catering to a diverse range of applications, including crowd-sourced mapping, among others. The leading approach for accomplishing 3D reconstruction is the *Structure-from-Motion* (SfM) algorithm, tasked with simultaneously estimating camera parameters and generating a 3D point cloud (Ullman, 1979). This research domain is primarily divided into two categories: *Incremental* (Heinly et al., 2015; Schönberger & Frahm, 2016; Snavely et al., 2006; 2008; Wu, 2013) and *Global methods* (Cui & Tan, 2015; Zhu et al., 2018; Sweeney; Pan et al., 2024).

Incremental algorithms carefully integrate images into the 3D reconstruction, achieving accuracy through repeated numerical optimizations. Although capable of yielding highly accurate results, these approaches are computationally intensive due to the necessity for multiple bundle adjustment runs (Triggs et al., 2000), rendering them less effective for reconstructing large datasets. In contrast, global methods optimize the entire pose graph, contaminated by noise for all cameras, in a single run, thus providing a fast and scalable solution. Even though global methods are generally seen as slightly less accurate compared with incremental techniques (Cui et al., 2017), they offer promising directions for rapid 3D model reconstruction. For example, recent advancements, such as GLOMAP (Pan et al., 2024), have shown their potential for scalability and efficiency, achieving state-of-the-art run-time and accuracy that is comparable to or surpassing COLMAP (Schönberger & Frahm, 2016). This paper is dedicated to global strategies, specifically presenting an algorithm designed to enhance camera position estimation. This is achieved using a Transformer architecture-based technique to filter out outlier edges within the pose graph.

Global methods start by identifying image pairs that share a common field-of-view, utilizing image retrieval techniques like the visual bag-of-words algorithm (Filliat, 2007) or NetVLAD (Arandjelovic et al., 2016). Following this, relative pose estimation is conducted for these pairs through a robust estimation method, for instance, RANSAC (Fischler & Bolles, 1981) or one of its state-of-the-art variants (Barath et al.; 2022). With camera rotations established, the process estimates camera translations, keeping the rotations fixed, and generates a 3D point cloud. The final step involves applying bundle adjustment to enhance the precision of the 3D model and camera parameters.

Figure 1: **Learned outlier filtering** takes a pose graph as input, where nodes represent images and edges relative poses. To avoid memory issues, the graph is clustered into several subgraphs. Each is converted into a line graph, where images become edges and the edges images. Each edge in the line graph is equipped with the pre-estimated camera orientation and the embedding (Oquab et al., 2023) of the underlying image and each vertex with the estimated relative pose $(t_{ij}R_{ij})$. Finally, a Transformer network predicts inlier/outlier labels for the relative poses.

Translation averaging, alternatively known as synchronization, has a rich literature (Govindu, 2001; Jiang et al., 2013; Moulon et al., 2013; Wilson & Snavely, 2014; Tron & Vidal, 2009; Ozyesil & Singer, 2015; Arrigoni et al., 2015b; Zhuang et al., 2018) and stands as one of the most complex challenges within global pipelines. It is typically approached as an optimization that seeks consistency between estimated relative translations and the unknown global camera positions. The core difficulty stems from the absence of known translation scales, making it hard to distinguish between short and long translations based on angles alone. Noise significantly impacts the process, particularly affecting short edges and often rendering them outliers. Furthermore, estimating robust relative translations from view pairs is generally less stable than rotation estimation (Barath et al., 2022). These issues make filtering outlier translation edges a crucial step to simplify the translation averaging problem.

Translation filtering, in contrast, has not been as extensively explored as averaging. 1DSfM (Wilson & Snavely, 2014) introduced a theoretical method for discarding outlier edges, which, despite its solid foundation, has been found to yield only marginal benefits in practice (Manam & Govindu, 2024). Another recent notable effort (Manam & Govindu, 2024) shifts the focus from filtering outliers to eliminating camera configurations detrimental to translation averaging, specifically, configurations involving camera triplets representing quasi-linear motion.

The main contribution of this paper is a learning-based pipeline designed to identify outlier edges within a pose graph of relative translations. Our approach (visualized in Fig. 1) transforms the pose graph into a line graph, wherein cameras become edges and relative translations nodes. This setup frames outlier detection as a binary classification task, where each node is assessed as either an inlier or an outlier through a Transformer-based architecture. We introduce a graph clustering approach to address the increase in memory associated with transforming the pose graph into a line graph. This enables applying the proposed network to graphs of any size, facilitating scalability. Across a number of large-scale and real-world datasets, the proposed pose graph filtering method leads to substantial improvements in camera position estimation, compared to the baseline outlier filter, 1DSfM (Wilson & Snavely, 2014). When our method is combined with the recent degeneracy filter (Manam & Govindu, 2024), it leads to the best results in all tested accuracy metrics across all datasets.

## 2 RELATED WORK

**Global Structure-from-Motion** reconstructs the scene in 3D and obtains the global camera poses by integrating estimates of relative poses between pairs. Predominantly, this integration is achieved through subsequent rotation (Olsson & Enqvist, 2011; Moulon et al., 2013) and translation averaging (Zhuang et al., 2018), though some studies have opted to perform averaging within SE(3) (Cui & Tan, 2015). Following the determination of rotations, the method proceeds to ascertain translations and structural details that are, finally, jointly optimized by bundle adjustment. Several open-source frameworks support global Structure-from-Motion, including Theia (Sweeney) and OpenMVG (Moulon et al., 2016). We have incorporated our relative translation filtering method into

the Theia framework, keeping the rest of the pipeline unchanged. Nonetheless, our advancements are versatile and not restricted to this particular pipeline.

**Rotation averaging** has a long history in computer vision. Govindu (Govindu, 2001) linearizes the problem using a quaternion representation, while Martinec and Pajdla (Martinec & Pajdla, 2007) simplify it by omitting certain non-linear constraints. Wilson et al. (Wilson et al., 2016) examine the conditions that make the problem tractable. Eriksson et al. (Eriksson et al., 2018) solve it via strong duality. Semidefinite programming-based (SDP) relaxation approaches (Arie-Nachimson et al., 2012; Fredriksson & Olsson, 2012) provide optimality guarantees by minimizing the chordal distance (Hartley et al., 2013). Dellaert et al. (Dellaert et al., 2020) sequentially lift the problem into higher-dimensional rotations $SO(n)$ to avoid local minima where standard numerical optimization techniques may fail (Levenberg, 1944; Marquardt, 1963). To handle outliers in relative rotations, various robust loss functions have been investigated (Hartley et al., 2011; Chatterjee & Govindu, 2013; 2017; Sidhartha & Govindu, 2021; Zhang et al., 2023).

The objective of **translation averaging** is to determine the absolute camera positions by leveraging estimated pairwise unscaled relative translations. Govindu (Govindu, 2001) approaches this by minimizing the cross-product of input and derived directions from absolute translations. Jiang et al. (Jiang et al., 2013) utilize the geometric constraints of triangle formations among triplets of nodes to approach the problem. Moulon et al. (Moulon et al., 2013) suggest a solution involving the minimization of a relaxed problem via the $L_\infty$ norm. Wilson et al. (Wilson & Snavely, 2014) aim to reduce the discrepancy between the input directions and those deduced from absolute translations. Tron et al. (Tron & Vidal, 2009) focus on minimizing squared relative displacements in a distributed framework. Ozyesil et al. (Ozyesil & Singer, 2015) introduce the Least Unsquared Deviations (LUD) method to extend (Tron & Vidal, 2009), incorporating L1 loss for enhanced robustness, thus framing the problem within a convex program. Arrigoni et al. (Arrigoni et al., 2015b) aim at minimizing the squared error from the orthogonal projection of estimated relative translations onto the input directions. Similarly, Goldstein et al. (Goldstein et al., 2016) minimize orthogonal projections through ADMM, adopting L1 loss for robustness. Zhuang et al. (Zhuang et al., 2018) offer a relaxation of the cost metrics in (Wilson & Snavely, 2014) by aligning estimated relative translations with observed directions, naming it Bilinear Angle-based Translation Averaging (BATA). Additional methodologies include leveraging two-view and three-view camera geometry to frame the problem (Hartley & Zisserman, 2003; Arie-Nachimson et al., 2012; Moulon et al., 2013), determining edge scales via cycles in a network prior to solving for absolute translations (Arrigoni et al., 2015a; 2016), or employing point correspondence constraints (Cui et al., 2015; 2016), refining input directions iteratively (Manam & Govindu, 2022), averaging matrices from two-view geometries (Kasten et al., 2019a;b), and making use of the matrix structure resulting from pairwise displacements (Dong et al., 2020). While these algorithms are crucial to getting accurate camera positions, they often fail due to degeneracies and outliers in the estimated pose graph edges.

**Pose graph filtering** serves as a strategy for removing inaccurate relative poses, thereby eliminating outliers to enhance the robustness of pose averaging. Given that translation averaging tends to be more prone to noise and outliers than rotation estimation in practice, most filtering methods assume the rotations to be pre-estimated and focus on improving translations. Zach et al. (Zach et al., 2010) introduce a method to exclude outlier edges based on loop consistency. The 1DSfM approach (Wilson & Snavely, 2014) involves projecting relative translation directions onto randomly selected 3D directions and then evaluating discrepancies within this one-dimensional subspace. This random projection process is iterated multiple times to accumulate instances of inconsistency for each edge. Such accumulated inconsistency metrics are then employed to eliminate discordant edges across multiple random projections. Shen et al. (Shen et al., 2016) focus on identifying and retaining a subset of reliable edges to ensure strong connectivity among cameras. The more recent study by Manam et al. (Manam & Govindu, 2024) highlights that the main issue often lies not with the outlier edges themselves but with edges that contribute to forming skewed triangles, which can lead to degenerate conditions in translation averaging. Our method focuses on filtering outlier edges and can be straightforwardly combined with any other filters, such as (Manam & Govindu, 2024).

Recently, **learning-based approaches** have emerged, primarily to solve the rotation averaging problem. NeuRoRa (Purkait et al., 2020) introduces a dual-network system: one for cleaning the view graph of relative rotations and another for fine-tuning, which optimizes an initialization of absolute orientations derived from the cleaned graph in a single step. MSP (Yang et al., 2021) proposes an end-to-end framework that initializes and optimizes orientations using multiple measurements

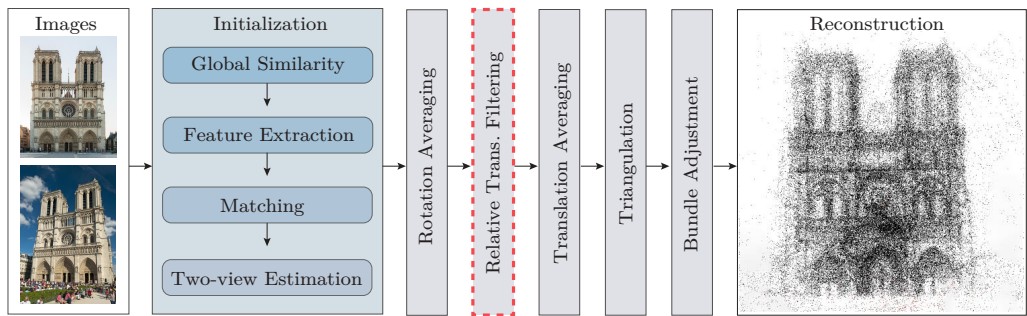

Figure 2: In **global Structure-from-Motion pipelines**, an image collection is fed into the initialization phase that selects potentially overlapping image pairs, performs feature detection and matching, and, finally, estimates relative poses. Next, rotation averaging runs to obtain the global camera orientations. Then the pose graph edges are filtered to remove outliers (e.g., by 1DSfM (Wilson & Snavely, 2014) or the proposed method) or degenerate configurations (Manam & Govindu, 2024). Translation averaging obtains the camera positions from the filtered graph with fixed orientations. Finally, the 3D structure is triangulated, and bundle adjustment optimizes all parameters jointly.

and incorporates neighborhood information. PoGO-Net (Li & Ling, 2021) presents a novel pose graph optimization (PGO) strategy implemented through a Graph Neural Network architecture, aiming to estimate absolute camera orientations accurately. DMF-synch (Tejus et al., 2023) utilizes a matrix factorization technique for pose extraction. While these methods show promising results, they primarily focus on rotation estimation. Contrarily, we argue that rotation averaging, in practice, tends to be significantly more accurate than estimating camera positions. Consequently, our work concentrates on enhancing translation averaging by filtering outlier relative translation edges.

## 3 FILTERING OUTLIER EDGES IN A POSE GRAPH

**Problem Statement.** Let us assume that we are given a set of images $\mathcal{I}$ and a pose graph $\mathcal{G} = (\mathcal{V}, \mathcal{E})$, with $\mathcal{V} \subseteq \mathcal{I}$ denoting the images within the graph and $\mathcal{E} = \{(v_i, v_j) \mid v_i, v_j \in \mathcal{V}\}$ comprising the edges. A mapping $T : \mathcal{E} \to \text{SE}(3)$ is defined such that $T(v_i, v_j) = (\mathbf{R}, \mathbf{t})$ represents the relative transformation from view $v_i$ to $v_j$, where $(v_i, v_j) \in \mathcal{E}$, $\mathbf{R} \in \text{SO}(3)$ is the relative rotation, and $\mathbf{t} \in \mathbb{R}^3$ is the relative translation, initially set to be unit-scaled ($\mathbf{t}^\mathrm{T}\mathbf{t} = 1$). The construction of graph $\mathcal{G}$, a preliminary step in both global and incremental SfM methods, involves image retrieval, feature detection and matching, and robust two-view geometry estimation (Schonberger & Frahm, 2016; Barath et al., 2021). An overview of global pipelines is depicted in Fig. 2.

Following the construction of the initial pose graph $\mathcal{G}$, the global SfM process is approximately the following. Initial steps include two-view geometry filtering, e.g., discarding edges with inlier counts below a threshold, such as 30 (i.e., the default in Theia (Sweeney)). Subsequently, rotation averaging, e.g., by using the robust method of Chatterjee et al. (Chatterjee & Govindu, 2013), determines the absolute camera orientations independently of positions. Rotation averaging generally achieves higher accuracy than position averaging, with errors typically within a few degrees (Li & Ling, 2021). Therefore, we assume known camera orientations $\mathbf{R}_v$ for all $v \in \mathcal{V}$ in the rest of the paper, concentrating on estimating the global camera translations $\mathbf{t}_v$. In global SfM, the step after rotation averaging usually involves re-estimating translations based on these orientations, filtering outlier edges to ensure consistency or eliminate degeneracy, e.g., by projecting onto 1D subspaces (Wilson & Snavely, 2014) or identifying degenerate motions (Manam & Govindu, 2024). The subsequent phases encompass translation averaging to derive global positions from the refined pose graph edges, triangulating to obtain a 3D point cloud, and executing bundle adjustment to jointly optimize all parameters by minimizing the re-projection error in pixels.

**Line Graph Representation.** In this paper, we aim to learn a filter $\theta : \mathcal{E} \to \{0, 1\}$ that identifies an edge (representing a relative translation estimate) as an inlier (1) or an outlier (0). To achieve this, we utilize a Graph Neural Network (GNN). While the direct learning of an edge classifier is feasible (Kim et al., 2019), we observed it to yield unstable results, often leading to excessive edge removal from

the pose graph and, consequently, significant loss of cameras crucial for the 3D reconstruction. To mitigate this issue, we propose converting the pose graph $\mathcal{G}$ into a line graph $L(\mathcal{G})$.

A line graph $L(\mathcal{G})$ is constructed from the original graph $\mathcal{G}$ by associating each vertex in $L(\mathcal{G})$ with an edge in $\mathcal{G}$. Specifically, if $\mathcal{G} = (\mathcal{V}, \mathcal{E})$, then in $L(\mathcal{G}) = (\mathcal{V}_L, \mathcal{E}_L)$, each edge $(v_i, v_j) \in \mathcal{E}$ becomes a vertex $v_{ij} \in \mathcal{E}_L$. Two vertices $v_{ij}$ and $v_{kl}$ in $L(\mathcal{G})$ are connected by an edge if and only if their corresponding edges in $\mathcal{G}$ share a common vertex, that is, $v_j = v_k$ or $v_i = v_l$. Mathematically speaking, the edge set $\mathcal{E}_L$ of $L(\mathcal{G})$ is defined as follows:

$$\mathcal{E}_L = \{(v_{ij}, v_{kl}) \mid (v_i = v_k \vee v_j = v_l) \wedge (v_i, v_j), (v_k, v_l) \in \mathcal{E}\}. \tag{1}$$

This transformation allows us to reformulate edge classification in $\mathcal{G}$ as vertex classification in $L(\mathcal{G})$, facilitating the application of vertex-focused GNN architectures for outlier detection.

**Outlier Filtering with a Graph Neural Network.** In the pose graph represented as a line graph $L(\mathcal{G})$, we attribute to each vertex – which corresponds to an edge in the original graph $\mathcal{G}$ – the estimated relative pose, consisting of a 3D rotation and unscaled translation. Note that these relative rotations are recalculated from the global camera orientations acquired earlier by rotation averaging. For each edge in $L(\mathcal{G})$, linked to vertex $v$ in $\mathcal{G}$, we assign two features concatenated into a single vector: (1) The 3D global orientation $\mathbf{R}_v$, and (2) the image embedding $\mathbf{d}_v \in \mathbb{R}^d$ corresponding to the input image $I_v \in \mathcal{I}$. We observed that leveraging context from the underlying image helps recognize outlier edges. We obtain embedding $\mathbf{d}_v$ using DINOv2 (Oquab et al., 2023) off-the-shelf, which yields a 384-dimensional image embedding. Consequently, the concatenated feature vector assigned to each edge in $L(\mathcal{G})$ is 393-dimensional. DINOv2 features capture the content of images, so neighboring images should have similar features, allowing the incorporation of such high-level information directly into the view graph. If two images are connected in the graph, we expect them to have overlapping views and, thus, similar DINOv2 features (Keetha et al., 2023).

Next, we will discuss the network structure. Our Graph Neural Network employs three `TransformerConv` layers with ReLU and dropout operations, the latter nullifying elements of the input node feature matrix with a 0.3 probability. `TransformerConv` is a Graph Convolutional Network (GCN) layer enhanced with a self-attention mechanism. It integrates edge attributes in both the computation of the attention coefficients and in the node update process. Operator `TransformerConv` is from (Shi et al., 2020) and is implemented in PyTorch Geometric Library. The node update mechanism, detailed in Eq. 2, combines the transformed representation of the current node $x_i$ with aggregated information from its neighboring nodes $j \in \mathcal{N}(i)$ and the connecting edges $e_{i,j}$. The contribution of each neighbor $x_j$ and edge $e_{i,j}$ to the updated state $x_i'$ of the node is modulated by the attention coefficient $\alpha_{i,j}$, calculated using a scaled dot-product attention mechanism. This attention framework, outlined in Eq. 3, leverages the query $W_3 x_i$, the key $W_4 x_j$, and the dimensionality of hidden channels $d$, with transformed edges $W_5 e_{i,j}$ enriching the key as follows:

$$x_i' = W_1 x_i + \sum_{j \in \mathcal{N}(i)} \alpha_{i,j}(W_2 x_j + W_6 e_{i,j}), \tag{2}$$

where the attention weights $\alpha_{i,j}$ are determined as follows:

$$\alpha_{i,j} = \text{softmax}\left(\frac{(W_3 x_i)^{\text{T}}(W_4 x_j + W_5 e_{i,j})}{\sqrt{d}}\right). \tag{3}$$

In the last layer, we employ a gated residual connection to prevent over-smoothing (Shi et al., 2020). A parameter $\beta$ is learned that controls how much of the previous and aggregated representations contribute to the final representation as follows:

$$x_i' = \beta_i W_1 x_i + (1 - \beta_i)\big(\underbrace{\sum_{j \in \mathcal{N}(i)} \alpha_{ij} W_2 x_j}_{m_i}\big),$$

where $\beta$ is calculated as follows:

$$\beta_i = \text{sigmoid}\left(w_6^{\text{T}}\left[W_1 x_i, m_i, W_2 x_i - m_i\right]\right).$$

**Graph Clustering.** Converting graph $\mathcal{G}$ to line graph $L(\mathcal{G})$ might require a significant amount of memory, depending on the connectivity of $\mathcal{G}$. Precisely, the number of edges in the line graph $|\mathcal{E}_L|$ is

quadratic in the node degree in the original graph $\mathcal{G}$ as:

$$|\mathcal{E}_L| = \frac{1}{2} \sum_{v \in \mathcal{V}} \deg(v)^2 - m, \tag{4}$$

where $m$ is the number of nodes in the line graph and $v$ are the vertices in the original graph. This property often leads to a memory explosion in a sufficiently large graph with well-connected images, preventing running the proposed classification network.

To overcome this issue, we employ a graph clustering technique (Chen et al., 2020) to reduce the size of the graph $\mathcal{G}$ before its conversion to the line graph $L(\mathcal{G})$. With this clustering, we can control the maximum number of edges in the line graphs and, thus, the maximum storage complexity of the proposed method. The approach consists of partitioning the graph $\mathcal{G}$ into $k$ clusters, $C_1, C_2, \ldots, C_k$, such that each cluster $C_i$ contains a subset of vertices from $\mathcal{V}$. This partitioning is designed to minimize the intra-cluster edge cut while maximizing the inter-cluster separation. Formally, the objective is to solve the optimization problem:

$$\min \sum_{i=1}^{k} \mathrm{cut}(C_i, \mathcal{V} \setminus C_i), \tag{5}$$

where $\mathrm{cut}(C_i, \mathcal{V} \setminus C_i)$ denotes the total weight of edges removed to separate cluster $C_i$ from the rest of the graph. The optimization finds a partitioning that balances the cluster sizes while minimizing the edge cuts across all clusters. We set the edge weights to equal the number of pose inliers found by RANSAC, which proved to be a good indicator of the edge quality. During inference, an edge may be part of multiple subgraphs, thus receiving more than one inlier/outlier vote. To make a final decision, majority voting is applied. The algorithm is further detailed in the Alg. 1 in the appendix.

The resulting clusters are then treated as super-vertices in a reduced graph $\mathcal{G}'$, with edges between super-vertices representing the connections between clusters in the original graph $\mathcal{G}$. The weight of an edge between two super-vertices in $\mathcal{G}'$ is determined by the sum of the weights of the edges between vertices in the corresponding clusters in $\mathcal{G}$. The reduced graph $\mathcal{G}'$ is then converted to a line graph $L(\mathcal{G}')$, significantly reducing the memory requirements compared to directly converting the original graph $\mathcal{G}$ to its line graph.

This clustering-based approach mitigates the memory explosion problem and preserves the essential topological and connectivity information from the original graph, enabling the effective application of the proposed classification network on large and densely connected graphs.

## 4 EXPERIMENTS

**Implementation Details.** Our global Structure-from-Motion (SfM) framework is implemented using the Theia library (Sweeney), using its default settings except for the deactivation of relative translation re-estimation, which was observed to impact final accuracy negatively. Relative pose estimation between image pairs is performed using LO$^+$-RANSAC (Lebeda et al., 2012). For computing accurate global camera orientations, we employ the rotation averaging method proposed by Chatterjee et al. (Chatterjee & Govindu, 2013). The revised LUD algorithm (Zhuang et al., 2018) is employed to derive global positions from relative translations. Optimization of all camera and point parameters is executed through the Levenberg-Marquardt algorithm (Levenberg, 1944), as implemented in Ceres (Agarwal & Mierle, 2012), minimizing reprojection errors in pixels. For calculating the error metrics, the reconstructions are aligned to reference data robustly by RANSAC.

**Baselines.** Our method is compared with various translation filters within the same framework, including: *No filter*: Translation averaging without translation filtering. *MSAC score*: To have a similar process as proposed in (Manam & Govindu, 2022), we assign a weight to each point correspondence by calculating the implied MSAC score from its Sampson error given the estimated relative pose. The weight of an edge is calculated as the sum of these weights, similar to what is done in (Manam & Govindu, 2022), as follows:

$$\text{score} = \sum_{\text{inliers}} \left( 1 - \frac{\text{Sampson Error}}{\text{RANSAC Threshold}} \right). \tag{6}$$

Table 1: Mean position errors and recall thresholded at 5 for reconstructions by Theia (Sweeney) on the **PhotoTourism dataset** (Snavely et al., 2006), using unfiltered relative translations (w/o Filter), MSAC score Manam & Govindu (2022), CleanNet (Purkait et al., 2020), the Triangle filter (Manam & Govindu, 2024), 1DSfM (Wilson & Snavely, 2014), 1DSfM combined with the Triangle filter, and our proposed method, both standalone and in conjunction with the Triangle filter. Additionally, the results of the Oracle filter, removing all outlier translations, are also presented. The best results are shown in bold, and the second-best ones are underlined.

| Scene | B. Museum | F. Cathedral | L. Memorial | M. Cathedral | M. Rushmore | Reichstag | Sacre Coeur | S. Familia | St. P. Cathedral | St. P. Square | AVG |
|---|---|---|---|---|---|---|---|---|---|---|---|
| Mean position error (↓) | | | | | | | | | | | |
| w/o Filter | 2.83 | 2.26 | **2.49** | **1.97** | **2.79** | 2.66 | 1.95 | 1.77 | 2.29 | 2.79 | 2.38 |
| MSAC score | 2.75 | 2.22 | 2.56 | 1.99 | **2.79** | 2.60 | 1.97 | 1.77 | 2.32 | **2.78** | 2.37 |
| CleanNet | 2.53 | 2.25 | 2.50 | 1.99 | **2.79** | 2.64 | 1.95 | 1.76 | **2.28** | 2.79 | 2.35 |
| Triangle | 2.83 | 2.26 | **2.49** | 2.01 | **2.79** | 2.66 | 1.94 | 1.78 | 2.29 | 2.79 | 2.38 |
| 1DSfM | **2.52** | 2.27 | 2.58 | 1.99 | **2.79** | 2.60 | 1.95 | 1.85 | 2.35 | 2.79 | 2.37 |
| 1DSfM+Tri. | 2.88 | 2.25 | 2.58 | 2.01 | **2.79** | 2.63 | 1.96 | 1.77 | 2.30 | 2.82 | 2.40 |
| Ours | **2.52** | 2.14 | **2.49** | 1.98 | **2.79** | 2.69 | **1.86** | **1.74** | 2.29 | 2.79 | 2.33 |
| Ours+Tri | **2.52** | **1.89** | **2.49** | 1.98 | **2.79** | 2.42 | **1.86** | **1.74** | 2.29 | 2.79 | **2.28** |
| Oracle | 2.15 | 1.19 | 1.22 | 1.87 | 2.47 | 2.17 | 1.35 | 1.35 | 1.99 | 2.38 | 1.81 |
| Position Recall@5 (↑) | | | | | | | | | | | |
| w/o Filter | 81.64 | 88.24 | 85.87 | 90.91 | **85.71** | 86.67 | 81.65 | 93.69 | **91.46** | 89.52 | 87.54 |
| MSAC score | 86.04 | 87.25 | **92.82** | 90.91 | 85.71 | 89.33 | 81.05 | 93.94 | 90.48 | 88.32 | 88.59 |
| CleanNet | 91.20 | 87.25 | 86.10 | 89.26 | 85.71 | 89.33 | 81.73 | 93.94 | **91.46** | 90.04 | 88.60 |
| Triangle | 83.61 | **89.22** | 92.70 | 90.08 | 85.71 | **90.67** | 81.82 | 93.94 | 90.97 | 89.64 | 88.84 |
| 1DSfM | **92.11** | **89.22** | 87.28 | **91.74** | 85.71 | 88.00 | 81.73 | 92.42 | 89.98 | 90.04 | 88.82 |
| 1DSfM+Tri. | 81.18 | 88.24 | 88.10 | 89.26 | 85.71 | 88.00 | 81.65 | 93.94 | 90.97 | 88.44 | 87.55 |
| Ours | 91.35 | **89.22** | 92.34 | 90.91 | 85.71 | 89.33 | 82.58 | **94.70** | 90.97 | **90.12** | **89.72** |
| Ours+Tri | 91.35 | 82.35 | 90.08 | 85.71 | 85.33 | 84.00 | **94.95** | 93.69 | 90.97 | **90.12** | 88.86 |
| Oracle | 94.23 | 98.04 | 98.94 | 93.39 | 84.96 | 90.67 | 91.76 | 95.71 | 92.45 | 93.08 | 93.32 |

Table 2: Mean position errors in meter and recall thresholded at 10 meter for reconstructions by Theia (Sweeney) on the **1DSfM dataset** (Wilson & Snavely, 2014), using unfiltered relative translations (w/o Filter), MSAC score Manam & Govindu (2022), CleanNet (Purkait et al., 2020), the Triangle filter (Manam & Govindu, 2024), 1DSfM (Wilson & Snavely, 2014), 1DSfM combined with the Triangle filter, and our proposed method, both standalone and in conjunction with the Triangle filter. Additionally, the results of the Oracle filter, removing all outlier translations, are also presented. On M. N. Dame, the MSAC score and our method could not be aligned with the COLMAP reconstruction for evaluation due to reconstructing a different set of cameras. The best results are shown in bold, and the second-best ones are underlined.

| Scene | Alamo | E. Isl. | Gendar. | M. Metro | M. N. Dame | NYC Lib. | N. Dame | P. del Pop. | Piccad. | R. Forum | T. of London | Trafal. | U. Square | V. Cath. | Yorkm. | AVG |
|---|---|---|---|---|---|---|---|---|---|---|---|---|---|---|---|---|
| Mean position error (↓) | | | | | | | | | | | | | | | | |
| w/o Filter | 10.99 | 96.48 | 46.44 | 30.61 | 8.52 | 17.43 | 5.44 | 20.38 | 17.45 | 25.39 | 53.18 | 1749.00 | 29.68 | 18.14 | 39.47 | 154.29 |
| MSAC score | 11.07 | 96.56 | 48.95 | 30.10 | – | 21.54 | 5.46 | 19.18 | 17.01 | 25.57 | 99.22 | 1762.31 | 30.30 | 18.24 | 38.18 | 158.83 |
| CleanNet | 10.86 | 96.84 | 54.48 | 29.94 | 8.61 | 16.29 | 5.45 | 19.10 | 17.16 | 25.50 | 48.51 | 1763.33 | 30.41 | 17.74 | 42.43 | 155.57 |
| Triangle | 6.66 | **19.82** | 46.38 | 21.75 | **6.08** | 6.36 | 5.45 | **18.56** | 11.06 | 22.15 | 36.57 | 78.22 | 14.95 | 13.86 | 14.78 | 22.61 |
| 1DSfM | 11.02 | 96.46 | 49.87 | 30.28 | 11.10 | 17.36 | 5.46 | 19.11 | 17.73 | 27.53 | 55.97 | 1742.24 | 29.71 | 18.86 | 37.28 | 154.21 |
| 1DSfM+Tri. | 6.77 | 19.87 | 51.18 | 23.02 | 9.77 | **6.01** | 5.32 | 19.26 | **9.69** | 23.95 | 41.78 | 75.03 | 14.75 | **13.74** | 16.34 | 23.34 |
| Ours | 10.80 | 23.69 | **42.89** | 27.80 | – | 10.81 | 5.44 | 19.00 | 16.90 | 23.57 | 39.80 | **73.07** | 27.43 | 16.29 | 40.03 | 26.97 |
| Ours+Tri | **6.62** | 21.56 | 43.85 | **19.95** | – | 6.06 | **5.29** | 19.16 | 10.90 | **21.18** | **29.08** | 73.29 | **14.18** | 14.02 | **10.95** | **21.15** |
| Oracle | 7.69 | 20.91 | 41.38 | 23.24 | 6.37 | 5.25 | 4.83 | 17.77 | 12.49 | 18.54 | 30.43 | 2082.17 | 17.08 | 13.06 | 10.83 | 164.69 |
| Position Recall@10m (↑) | | | | | | | | | | | | | | | | |
| w/o Filter | 65.18 | 37.28 | 13.53 | 25.12 | **77.73** | 49.63 | **87.43** | 37.38 | 64.51 | **35.29** | 19.90 | 13.04 | 40.22 | 41.65 | 23.83 | 39.57 |
| MSAC score | 65.05 | 37.28 | **21.24** | 23.24 | – | 52.58 | **87.43** | 37.74 | 64.46 | 34.46 | 4.48 | 14.28 | 37.29 | 39.59 | 23.47 | 38.77 |
| CleanNet | 66.36 | 38.70 | 15.93 | 25.12 | 75.30 | 52.33 | 87.13 | 37.62 | **65.29** | 34.23 | 19.07 | 6.96 | 37.60 | 41.75 | 25.27 | 39.53 |
| Triangle | **69.78** | **44.39** | 7.96 | 28.64 | 71.05 | **75.18** | 87.06 | 38.09 | 37.76 | 23.05 | 14.76 | **41.91** | 42.27 | **49.10** | 31.70 | 44.18 |
| 1DSfM | 65.44 | 37.28 | 17.32 | 23.71 | 64.57 | 49.63 | 87.13 | 37.38 | 63.73 | 28.45 | 17.25 | 5.41 | 41.76 | 41.03 | 23.10 | 38.47 |
| 1DSfM+Tri. | 68.46 | 42.81 | 18.84 | 28.17 | 69.64 | 72.24 | 87.06 | 37.50 | 61.94 | 30.93 | 14.93 | 15.15 | **42.99** | 41.24 | 43.68 | 43.28 |
| Ours | 64.91 | 41.55 | 14.16 | 25.59 | – | 60.44 | 87.06 | **38.92** | 63.58 | 32.06 | 20.9 | 15.27 | 40.06 | 35.26 | 28.88 | 40.78 |
| Ours+Tri. | 68.46 | 36.65 | 18.33 | **29.11** | – | 70.76 | 87.21 | 35.14 | 63.58 | 33.33 | **25.04** | 15.49 | 42.37 | 42.27 | **54.15** | **44.42** |
| Oracle | 76.74 | 45.34 | 28.45 | 38.03 | 79.55 | 88.94 | 89.39 | 50.12 | 80.51 | 48.87 | 34.83 | 8.93 | 61.17 | 59.69 | 77.98 | 56.36 |

We filter edges by removing those with a score lower than 30. Note that we cannot use directly the weight from (Manam & Govindu, 2022) as they need the absolute camera positions. *1DSfM*: Utilizes the filter from (Wilson & Snavely, 2014) to eliminate incorrect relative translations through projections into random 1D subspaces. *Triangle*: Employs a filter from (Manam & Govindu, 2024) targeting the removal of degenerate camera triplet configurations rather than outliers. Skewed triangles, where the minimal angle falls below a threshold, are removed from the pose graph. This method can be straightforwardly combined with the proposed one, as we will demonstrate with experiments. *CleanNet*: An outlier detection network that filters outliers in the initial stage of the

Table 3: Mean position errors in meter and recall thresholded at 1 meter for reconstructions by Theia (Sweeney) on the **ScanNet dataset** (Dai et al., 2017), using unfiltered relative translations (w/o Filter), MSAC score Manam & Govindu (2022), CleanNet (Purkait et al., 2020), the Triangle filter (Manam & Govindu, 2024), 1DSfM (Wilson & Snavely, 2014), 1DSfM combined with the Triangle filter, and our proposed method, both standalone and in conjunction with the Triangle filter. Additionally, the results of the Oracle filter, removing all outlier translations, are also presented. The best results are shown in bold, and the second-best ones are underlined. On Scene 0207, Oracle reconstructs a different set of cameras than the proposed filter; thus, we exclude this scene from the average recall and report the unnormalized recall on it.

| Scene | 0000 | 0059 | 0106 | 0169 | 0181 | 0207 | AVG |
|---|---|---|---|---|---|---|---|
| | | | Mean position error (↓) | | | | |
| w/o Filter | 0.65 | **0.58** | 0.87 | 0.58 | 1.21 | 0.81 | 0.78 |
| MSAC score | **0.62** | 1.20 | 1.03 | 1.27 | 1.26 | 1.13 | 1.09 |
| CleanNet | 0.65 | **0.58** | 0.87 | 0.58 | 1.21 | 0.81 | 0.78 |
| Triangle | 0.65 | 0.60 | 0.87 | **0.48** | 1.17 | 0.59 | 0.73 |
| 1DSfM | 0.69 | **0.58** | 1.08 | 0.55 | 1.26 | 0.80 | 0.83 |
| 1DSfM+Tri. | 0.69 | 0.59 | 0.90 | **0.48** | **0.70** | 0.61 | 0.66 |
| Ours | 0.66 | **0.58** | 0.85 | 0.63 | 1.12 | 0.55 | 0.73 |
| Ours+Tri. | 0.66 | 0.60 | **0.57** | 0.50 | **1.02** | **0.36** | **0.62** |
| Oracle | 0.49 | 0.61 | 0.39 | 0.45 | 0.64 | 0.37 | 0.49 |
| | | | Position Recall@1m (↑) | | | | |
| w/o Filter | 90.30 | 89.06 | 91.05 | 92.81 | 24.65 | 58.88 | 77.57 |
| MSAC score | **94.12** | 64.33 | 54.98 | 53.14 | 36.27 | 47.38 | 60.57 |
| CleanNet | 90.20 | 89.06 | 91.05 | 92.97 | 24.65 | 58.71 | 77.59 |
| Triangle | 90.24 | 89.06 | 90.94 | 93.71 | 28.03 | 97.97 | 78.40 |
| 1DSfM | 82.66 | **90.35** | 72.87 | **93.93** | 20.45 | 58.38 | 72.05 |
| 1DSfM+Tri. | 82.18 | 89.47 | 85.85 | 92.91 | **75.52** | 96.79 | 85.19 |
| Ours | 87.43 | 88.95 | 84.70 | 91.10 | 42.87 | 95.64 | 79.01 |
| Ours+Tri. | 86.86 | 88.77 | **94.12** | 92.43 | 66.36 | **99.49** | **85.71** |
| Oracle | 90.13 | 90.94 | 99.38 | 94.41 | 86.07 | 100.00 | 92.19 |

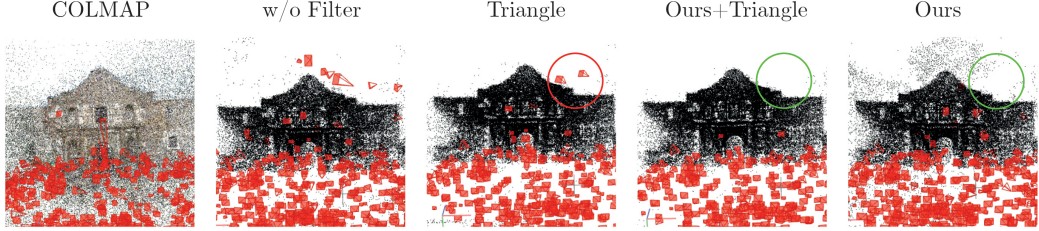

| COLMAP | w/o Filter | Triangle | Ours+Triangle | Ours |

Figure 3: Reconstructions of scene Alamo from the 1DSfM dataset obtained by Theia's Global SfM pipeline (Sweeney) with different filtering methods. An example where only the proposed method can remove incorrect cameras can be seen on the right side of the building.

NeuRoRA framework (Purkait et al., 2020) by assessing relative rotations. *Oracle*: Demonstrates the potential maximum accuracy of an ideal outlier filter by excluding relative translations with errors exceeding $20°$ w.r.t. ground truth. Please note that while the accuracy of Oracle may be the upper bound without outliers, it can be surpassed by removing degenerate configurations.

**Datasets.** Evaluations were conducted on the 1DSfM (Wilson & Snavely, 2014), Photo-Tourism (Snavely et al., 2006), and ScanNet (Dai et al., 2017) datasets. The 1DSfM dataset, encompassing 15 landmark scenes with internet-sourced photos, includes two-view matches, epipolar geometries, and a reference incremental SfM reconstruction (via Bundler (Snavely et al., 2006; 2008)) for error analysis. Given that Bundler is nowadays considered outdated, we generated new reference reconstructions by COLMAP (Schonberger & Frahm, 2016). To ensure that the reconstruction is approximately metric, we robustly align it with the provided Bundler reconstruction. The scenes used for the PhotoTourism dataset are based on the CVPR Image Matching Challenge 2020. These scenes are non-metric, thus, the reported position errors are *not* in meters. The ScanNet dataset (Dai et al., 2017) consists of 1613 monocular sequences with ground truth poses and metric depth. To evaluate relative translation filters, we utilize the same six sequences as what Zhu et al. (Zhu et al., 2022) use.

**Training.** We trained the proposed method on scene Piazza San Marco from the PhotoTourism dataset, comprising 249 images and 10295 view graph edges in total. We use the COLMAP reconstruction to provide target inlier/outlier labels. We label a relative translation outlier if its error is higher than $20°$,

leading to 4418 outlier edges and to good performance in our experiments. As the validation set, we split scene Taj Mahal. We exclude these scenes from the main experiments. The model is trained for 300 epochs, with 0.009 learning rate and Binary Cross Entropy Loss, $5 * 10^{-4}$ weight decay, and Adam optimizer (Kingma & Ba, 2014) on 3 subgraphs, shuffled during training. We use this model in all experiments and on all datasets, demonstrating the generalization capabilities of the proposed method. Specifically, we train it on a *single* outdoor scene and test it on three large-scale datasets with significant domain gaps (indoor/outdoor), showing that it performs accurately across different noise and outlier distributions. For a fair comparison, we performed hyper-parameter tuning for the baselines on the same training data.

**Metrics** include mean position errors, recall rates at 1, 5 and 10. We report the mean error as it clearly shows large failures in the reconstruction. We include recall as a robust metric to show the accuracy of the well-reconstructed cameras. Other metrics are reported in the supplementary material. To make the recall rates fair across methods, we take the reconstructed cameras after applying the Oracle filter and calculate the accuracy on these cameras. For each camera missing from the reconstruction with a particular filter, we consider the error to be infinity for recall calculation. On 1DSfM and ScanNet, the errors are in meters, as the reference is a metric reconstruction. There, we report the recall thresholded at 10 and 1 meters. On PhotoTourism, the errors have no units. Thus, we report the recall at 5, which we chose so that the results are meaningful.

**PhotoTourism.** The results on the PhotoTourism dataset are shown in Table 1. MSAC score, CleanNet, the standalone Triangle, the 1DSfM filters, and their combination have a negligible impact. On average, upon integration with the Triangle filter, the proposed method outperforms the baselines in all accuracy metrics. This clearly shows that the two filters complement each other, the proposed one removes outliers, and the Triangle filter gets rid of the degenerate configurations. Employed independently of the Triangle filter, our approach secures best or second-best results. As expected, the Oracle filter achieves superior performance on nearly all scenes. However, it is crucial to highlight that its excellence is in outlier removal and does not extend to identifying degenerate configurations, which could adversely impact reconstruction quality. Moreover, the threshold of $20°$ employed for removing incorrect translations may not be optimal in all scenes. These are the reasons why the Oracle filter does not always lead to the best reconstructions.

**1DSfM.** Results on the 1DSfM dataset are shown in Table 2. Consistent with observations on the PhotoTourism dataset, the proposed filtering significantly surpasses the conventional outlier filtering technique, 1DSfM, in performance. Notably, it reduces the average position error by approximately a factor of 5 and enhances the recall rate by approximately 6%. The MSAC score baseline has significantly higher error implying that it fails to filter incorrect pose graph edges effectively. CleanNet only has a minor impact on accuracy. The combined Ours+Triangle method attains the highest recall and lowest mean errors.

In the reconstructions depicted in Fig. 3, COLMAP is used as the benchmark. The COLMAP reconstruction shows no cameras floating above the building. Only our method, alone and combined with the Triangle filter, avoids this issue, aligning with the cameras observed in COLMAP.

**ScanNet.** The performance on six scenes from ScanNet (Dai et al., 2017), as selected by (Zhu et al., 2022), is reported in Table 3. The trends are similar to those observed on other datasets. MSAC score is not improving over the baselines. Notably, CleanNet exhibits negligible enhancement over the scenario without any filter applied. Interestingly, the conventional outlier filter, 1DSfM, decreases the accuracy on average. However, combined with the Triangle filter, it enhances accuracy across all metrics. The proposed filter, on the other hand, improves performance across all accuracy metrics compared to the unfiltered approach, even without the Triangle filter. The proposed method achieves a recall rate that is approximately 10% higher than 1DSfM. When integrated with the Triangle filter, it attains the lowest mean errors and the highest recalls.

**Runtime.** The average run-time of the proposed method is 8.6 mins. for the PhotoTourism dataset, 2.55 mins. for the 1DSfM dataset, and 4.42 mins. for the ScanNet dataset. The running times on all scenes for all baselines are detailed in Table 8 in the supplement. Although filters like 1DSfM (Wilson & Snavely, 2014) are faster (1.8 secs. PhotoTourism, 0.6 secs. 1DSfM, 1.2 secs. ScanNet), it is crucial to emphasize that none of the filtering methods, including 1DSfM filtering, nor SfM itself are designed for real-time performance. Despite this, the global pipeline using Theia combined with our filtering method remains orders-of-magnitude faster than an incremental approach like

Table 4: Ablation study averaged over scenes T. of London, M. Rushmore, and E. Island. We report the mean position errors, the recall at 10, and the number of cameras.

| | Mean Position Error (↓) | Recall@10 (↑) | # of cameras (↑) |
|---|---|---|---|
| 1DSfM | 51.74 | 48.75 | **740** |
| Ours w/o image features | 37.78 | 46.14 | 464 |
| Ours w/ 2 layers | 24.39 | 47.33 | 565 |
| Ours trained with 8 clusters | 22.49 | 48.14 | 441 |
| Ours inference w/o clustering | 46.60 | 44.07 | 686 |
| Proposed | **22.09** | **51.39** | 548 |

COLMAP. For instance, on 11 scenes of the 1DSfM dataset, Theia takes an average of 23 minutes for reconstruction, compared to COLMAP's 886 minutes (Table 9 in the supplement). Thus, spending a few additional minutes on filtering is inconsequential in a SfM pipeline.

**Camera Numbers.** The average number of cameras retained across the tested datasets for each method is as follows: w/o Filter (1618), MSAC score (1612), CleanNet (1590), Triangle (1356), 1DSfM (1617), 1DSfM + Triangle (1318), Ours (1463), Ours + Triangle (1263), and Oracle (1341). When combined with the Triangle filter, the proposed method retains fewer cameras than the Oracle method. However, it is important to note that the reported recalls were calculated on the same set of cameras returned by the Oracle filter. The combination of our method with the Triangle filter achieves the best recalls while retaining the fewest cameras, indicating its effectiveness in removing inaccurate poses from the graph – specifically, cameras that could not be recovered accurately.

**Ablation studies.** To gain a more nuanced understanding of the proposed filtering, we run the following configurations on scenes T. of London, M. Rushmore, and E. Island, as presented in Table 4. *Ours w/o image features* removes the image embeddings from the network. Only two layers in the graph neural network instead of three are used in *Ours w/ 2 layers*. The next ablation explores how changing the number of clusters during training affects performance. In *Ours trained with 8 clusters*, we increase the number of clusters from the usual three to eight. In *Ours inference w/o clusters*, we do not employ our clustering method, so the entire graphs are used for inference. Additionally, we show the results of the entire pipeline with 1DSfM filtering as baseline. In Table 10 of the appendix, we provide additional ablations showing the importance of the network inputs.

Even without image features, the proposed method improves the mean position error upon the 1DSfM baseline. Reducing the number of layers has a minor effect on the accuracy. Artificially increasing the number of clusters during training leads to high camera loss due to splitting the graph into too small subgraphs. *Ours inference w/o clustering* runs the method on entire graphs to evaluate whether clustering introduces any approximation or loss of information. While it reconstructs the most cameras after the baseline, it has lower recall and a higher mean error compared to our proposed method. This shows that clustering followed by majority voting does not degrade performance, but improves accuracy across all metrics by focusing on local subgraphs to detect outlier edges.

## 5 CONCLUSION

This paper presents a novel filtering method that enhances camera position estimation in global SfM, demonstrating improved accuracy across a diverse set of datasets. By jointly addressing outliers with the proposed method and degenerate configurations by (Manam & Govindu, 2024), our approach ensures superior reconstruction quality while only being marginally slower than other alternatives. The proposed method without (Manam & Govindu, 2024) is superior, in terms of accuracy, to the standard outlier filtering techniques, e.g., 1DSfM (Wilson & Snavely, 2014). These results highlight the critical role of advanced filtering in Structure-from-Motion. The source code will be made public.

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

## A   APPENDIX

Next, we will provide additional metrics on the tested datasets and the processing times of all methods, a complexity analysis, additional ablation studies on the network inputs, more description on the clustering and visualizations.

## B   ADDITIONAL METRICS

The median position error and the number of reconstructed cameras using Theia (Sweeney) across all datasets are presented in Tables 5, 6, and 7. Consistent with the mean position error and recall rates, our method combined with the Triangle filter achieves the lowest median errors across all datasets. Notably, our proposed method consistently outperforms the 1DSfM baseline. The proposed method results in a similar number of reconstructed cameras to the baselines. However, combining it with the Triangle filter results in a reduction in the number of reconstructed cameras. It is important to highlight that the highest recall values achieved by the Ours+Triangle method (in the main paper) indicate that it only removes cameras that could not be reconstructed accurately.

To achieve a clearer understanding of the removed and kept translation directions, we present the angular error distribution for inliers and outliers as labeled by our proposed method or 1DSfM in Fig. 4 for three scenes of the 1DSfM dataset. It can be seen that the outliers removed by the proposed filter are mostly located in the right part of the histograms, indicating that the removed edges indeed have high errors. The 1DSfM filter, on the other hand, barely removes any edges. For example, 1DSfM does not remove anything in the Milan Cathedral scene. While the removed edges usually have high errors, such a minimal filtering has negligible impact on the final accuracy, as can be seen in the tables of the main paper.

On average, the relative error in degrees is significantly lower for inlier edges (37.69° 1DSfM, 43.19° PhotoTourism) compared to outlier edges (43.49° 1DSfM, 50.56° PhotoTourism) demonstrating that the proposed method effectively filters out less accurate edges and retains relative translations with reduced angular error. Across all six scenes analyzed, the distribution shows that inliers are more concentrated in regions with lower angular error, while outliers are more frequently observed in regions with higher angular error. This distinction underscores the method's capability to discriminate between potential inliers and outliers.

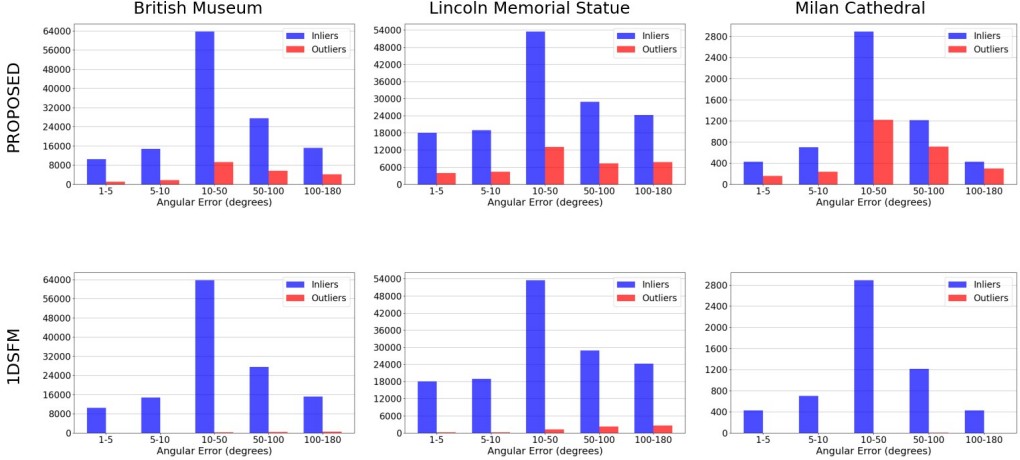

Figure 4: Relative error distribution of different scenes filtered with our proposed method and 1DSfM filter. Shown here are scenes British Museum, Lincoln Memorial Statue and Milan Cathedral from the PhotoTourism dataset.

Table 6: Median position errors and the number of cameras reconstructed by Theia (Sweeney) on the **1DSfM dataset** (Wilson & Snavely, 2014), using unfiltered relative translations (w/o Filter), MSAC score Manam & Govindu (2022), CleanNet (Purkait et al., 2020), the Triangle filter (Manam & Govindu, 2024), 1DSfM (Wilson & Snavely, 2014), 1DSfM combined with the Triangle filter, and our proposed method, both standalone and in conjunction with the Triangle filter. Additionally, the results of the Oracle filter, removing all outlier translations, are also presented. The best results are shown in bold, and the second-best ones are underlined.

| Scene | Alamo | E. Island | Gendarm. | M. Metro | M. N. Dame | NYC Library | N. Dame | P. del Popolo | Piccad. | R. Forum | T. of London | Trafalgar | U. Square | V. Cath. | Yorkm. | AVG |
|---|---|---|---|---|---|---|---|---|---|---|---|---|---|---|---|---|
| | | | | | Median position error (↓) | | | | | | | | | | | |
| w/o Filter | 5.38 | 16.10 | 41.38 | 21.79 | 5.30 | 10.97 | 2.43 | 14.66 | 6.83 | 14.71 | 31.81 | 58.64 | 12.78 | 12.90 | 22.44 | 19.49 |
| MSAC score | 5.67 | 16.12 | **30.49** | 22.56 | – | 9.48 | 2.41 | 14.03 | 6.87 | 15.23 | 84.55 | 60.32 | 12.95 | 13.95 | 23.15 | 22.70 |
| CleanNet | 5.22 | 16.61 | 50.56 | 22.52 | 5.11 | 9.74 | 2.44 | 14.02 | 6.74 | 14.61 | 32.37 | **41.30** | 13.22 | 12.31 | 21.69 | 18.81 |
| Triangle | **3.34** | **7.65** | 39.85 | 13.77 | 4.73 | 4.13 | **2.30** | 13.51 | **5.55** | 13.59 | 25.19 | 59.38 | 10.04 | 10.96 | 9.36 | 15.62 |
| 1DSfM | 5.25 | 16.29 | 45.76 | 22.09 | 6.74 | 11.12 | 2.44 | 14.29 | 7.08 | 17.02 | 34.42 | 64.41 | 12.17 | 12.64 | 25.22 | 20.73 |
| 1DSfM+Tri. | 3.40 | 8.94 | 43.76 | 16.67 | 5.97 | **3.74** | 2.38 | 13.13 | 5.96 | 15.57 | 29.06 | 56.69 | 9.79 | 11.24 | 10.52 | 16.49 |
| Ours | 5.27 | 10.23 | 36.93 | 20.37 | – | 6.97 | 2.35 | 12.98 | 6.58 | 14.00 | 27.41 | 55.72 | 12.12 | 12.25 | 20.34 | 17.39 |
| Ours+Tri. | 3.38 | 9.19 | 36.95 | **12.57** | – | 3.92 | **2.30** | **12.60** | 5.61 | **12.81** | 18.18 | 54.58 | 9.72 | 10.66 | **6.65** | **14.22** |
| Oracle | 3.77 | 13.28 | 31.74 | 14.78 | 4.54 | 2.99 | 2.02 | 9.98 | 3.94 | 10.37 | 16.46 | 36.68 | 7.50 | 7.96 | 5.51 | 11.93 |
| | | | | | # of cameras (↑) | | | | | | | | | | | |
| w/o Filter | **943** | **1229** | **1093** | 751 | **1507** | **1135** | **1422** | **1095** | **3615** | **1678** | 872 | **5092** | 1293 | **1704** | **645** | **1605** |
| MSAC score | 938 | 1212 | 1091 | 745 | – | 1123 | 1419 | 1078 | 3586 | 1677 | 863 | 5025 | 1251 | 1675 | **645** | 1595 |
| CleanNet | 918 | 1173 | 1048 | 681 | 1460 | 1018 | **1422** | 1047 | 3496 | 1635 | 813 | 4896 | 1174 | 1509 | 621 | 1527 |
| Triangle | 689 | 703 | 853 | 394 | 469 | 411 | 1400 | 836 | 2810 | 1378 | 603 | 3472 | 760 | 982 | 322 | 1072 |
| 1DSfM | **943** | **1229** | **1093** | 751 | **1507** | **1135** | **1422** | **1095** | **3615** | **1678** | 854 | 5072 | 1291 | **1704** | **645** | 1602 |
| 1DSfM+Tri. | 689 | 703 | 852 | 393 | 469 | 410 | 1396 | 831 | 2779 | 1370 | 607 | 3408 | 741 | 979 | 321 | 1063 |
| Ours | 929 | 761 | 970 | 675 | – | 886 | 1416 | 1051 | 3407 | 1522 | 744 | 3989 | 1268 | 1490 | 563 | 1405 |
| Ours+Tri. | 684 | 586 | 853 | 328 | – | 403 | 1391 | 753 | 2692 | 1160 | 534 | 3123 | 731 | 944 | 265 | 1032 |
| Oracle | 789 | 795 | 817 | 534 | 1435 | 487 | 1355 | 870 | 2959 | 1500 | 618 | 4256 | 691 | 1474 | 277 | 1248 |

Table 5: Median position errors and the number of cameras reconstructed by Theia (Sweeney) on the **PhotoTourism dataset** (Snavely et al., 2006), using unfiltered relative translations (w/o Filter), MSAC score Manam & Govindu (2022), CleanNet (Purkait et al., 2020), the Triangle filter (Manam & Govindu, 2024), 1DSfM (Wilson & Snavely, 2014), 1DSfM combined with the Triangle filter, and our proposed method, both standalone and in conjunction with the Triangle filter. Additionally, the results of the Oracle filter, removing all outlier translations, are also presented. The best results are shown in bold, and the second-best ones are underlined.

| Scene | B. Museum | F. Cathedral | L. Memorial | M. Cathedral | M. Rushmore | Reichstag | Sacre Coeur | S. Familia | St. P. Cathedral | St. P. Square | AVG |
|---|---|---|---|---|---|---|---|---|---|---|---|
| | | | | | Median position error (↓) | | | | | | |
| w/o Filter | 2.03 | 1.42 | 2.18 | **1.16** | 1.29 | **1.37** | 0.53 | 1.07 | 1.10 | **1.61** | 1.38 |
| MSAC score | **1.96** | 1.48 | 2.27 | 1.20 | 1.28 | 1.49 | 0.54 | 1.14 | **1.09** | 1.71 | 1.42 |
| CleanNet | 2.03 | 1.48 | 2.18 | 1.23 | 1.23 | 1.39 | 0.53 | 1.10 | 1.13 | 1.65 | 1.40 |
| Triangle | 2.03 | 1.48 | 2.18 | 1.22 | 1.23 | **1.37** | 0.54 | **1.06** | 1.10 | **1.61** | 1.38 |
| 1DSfM | 2.04 | 1.49 | **2.09** | 1.23 | 1.24 | 1.40 | 0.54 | **1.06** | **1.07** | 1.64 | 1.38 |
| 1DSfM+Tri.. | 2.03 | 1.46 | 2.12 | 1.20 | 1.24 | 1.45 | 0.54 | **1.06** | 1.09 | 1.71 | 1.39 |
| Ours | 2.01 | 1.18 | 2.14 | **1.22** | **1.22** | 1.52 | 0.51 | 1.15 | **1.09** | 1.66 | 1.37 |
| Ours+Tri. | 2.01 | **0.98** | 2.14 | **1.22** | **1.22** | 1.39 | **0.51** | 1.15 | **1.09** | 1.66 | **1.34** |
| Oracle | 1.67 | 0.66 | 0.87 | 1.09 | 0.78 | 1.13 | 0.35 | 0.75 | 0.95 | 1.37 | 0.96 |
| | | | | | # of cameras (↑) | | | | | | |
| w/o Filter | **660** | **108** | **850** | **123** | **138** | **75** | **1177** | **401** | **612** | **2503** | **665** |
| MSAC score | **660** | **108** | **850** | **123** | **138** | **75** | **1177** | **401** | **612** | **2503** | **665** |
| CleanNet | **660** | **108** | **850** | **123** | **138** | **75** | **1177** | **401** | **612** | **2503** | **665** |
| Triangle | **660** | **108** | **850** | 122 | **138** | **75** | **1177** | **401** | **612** | **2503** | **665** |
| 1DSfM | **660** | **108** | **850** | **123** | **138** | **75** | **1177** | **401** | **612** | **2503** | **665** |
| 1DSfM+Tri. | **660** | **108** | **850** | **123** | **138** | **75** | **1177** | **401** | **612** | **2503** | **665** |
| Ours | **660** | 107 | **850** | **123** | **138** | **75** | **1177** | **401** | **612** | **2503** | **665** |
| Ours+Tri. | **660** | 96 | **850** | **123** | **138** | **75** | **1177** | **401** | **612** | **2503** | 664 |
| Oracle | 659 | 102 | 849 | 121 | 133 | 75 | 1177 | 396 | 609 | 2501 | 662 |

## C   PROCESSING TIME

The runtime of all filtering methods applied to the datasets PhotoTourism, 1DSfM, and ScanNet is detailed in Table 8. Among these methods, 1DSfM is the most efficient, closely followed by CleanNet (Purkait et al., 2020), MSAC score based on (Manam & Govindu, 2022) and Triangle, the latter being a brute force implementation in C++ based on (Manam & Govindu, 2024). The Oracle filter also demonstrates speed, implemented in Python. It compares each edge in the view graph against the relative translation obtained from COLMAP. When our method is run prior to applying the Triangle filter, the Triangle filter is more efficient on many scenes, as it needs to iterate through fewer

Table 7: Median position errors and the number of cameras reconstructed by Theia (Sweeney) on the **ScanNet dataset** (Dai et al., 2017), using unfiltered relative translations (w/o Filter), MSAC score Manam & Govindu (2022), CleanNet (Purkait et al., 2020), the Triangle filter (Manam & Govindu, 2024), 1DSfM (Wilson & Snavely, 2014), 1DSfM combined with the Triangle filter, and our proposed method, both standalone and in conjunction with the Triangle filter. Additionally, the results of the Oracle filter, removing all outlier translations, are also presented. The best results are shown in bold, and the second-best ones are underlined.

| Scene | 0000 | 0059 | 0106 | 0169 | 0181 | 0207 | AVG |
|---|---|---|---|---|---|---|---|
| | | | Median position error (↓) | | | | |
| w/o Filter | 0.60 | **0.49** | 0.77 | 0.45 | 1.23 | 0.79 | 0.72 |
| MSAC score | 0.62 | 1.25 | 0.98 | 1.40 | 1.23 | 1.15 | 1.10 |
| CleanNet | **0.59** | 0.50 | 0.77 | 0.45 | 1.23 | 0.79 | 0.72 |
| Triangle | **0.59** | 0.52 | 0.76 | **0.36** | 1.18 | 0.58 | 0.66 |
| 1DSfM | 0.61 | **0.49** | 0.87 | 0.43 | 1.27 | 0.79 | 0.74 |
| 1DSfM+Tri. | 0.60 | 0.51 | 0.71 | 0.37 | **0.73** | 0.58 | 0.58 |
| Ours | **0.59** | 0.50 | 0.75 | 0.49 | 1.07 | 0.51 | 0.65 |
| Ours+Tri. | **0.59** | 0.53 | **0.52** | 0.38 | 0.87 | **0.30** | **0.53** |
| Oracle | 0.36 | 0.63 | 0.34 | 0.34 | 0.49 | 0.36 | 0.42 |
| | | | # of cameras (↑) | | | | |
| w/o Filter | **5572** | **1806** | **2259** | **2026** | **1885** | **1953** | **2584** |
| MSAC score | 5572 | 1806 | 2260 | 2022 | 1862 | 1941 | 2577 |
| CleanNet | 5571 | 1806 | **2259** | 2024 | 1876 | 1943 | 2580 |
| Triangle | 5560 | 1803 | 2256 | 1922 | 1686 | 760 | 2331 |
| 1DSfM | **5572** | **1806** | **2259** | **2026** | **1885** | **1953** | **2584** |
| 1DSfM+Tri. | 5537 | 1783 | 2234 | 1903 | 1172 | 741 | 2228 |
| Ours | 5571 | **1806** | 1911 | 2025 | 1498 | 1098 | 2318 |
| Ours+Tri. | 5542 | 1799 | 1546 | 1845 | 1238 | 588 | 2093 |
| Oracle | 5472 | 1730 | 1788 | 1891 | 1213 | 591 | 2114 |

edges compared to when it operates independently. As expected, the proposed filtering method is the slowest. It runs for 8.60 (PhotoTourism), 2.59 (1DSfM), and 4.42 (ScanNet) minutes on average. Let us note that this is still negligible compared with other components of a Structure-from-Motion pipeline, e.g., feature matching and final bundle adjustment. Moreover, our code can be further optimized by moving all its parts from Python to C++. Table 9 compares the runtime between Theia and COLMAP across 11 scenes from the 1DSfM dataset.

## D  COMPLEXITY ANALYSIS

The computational complexity of the method is described below and in algorithms 1 2 3 4.

**1. Graph clustering.** The graph clustering 1 has a time complexity of $\mathcal{O}(|\mathcal{E}|)$. The function for computing the $k$-way partition is implemented in COLMAP and utilizes the METIS library (Karypis & Kumar, 1997). It has a complexity of $\mathcal{O}(|\mathcal{E}|)$ as discussed in (Karypis & Kumar, 1997). Thus, the complexity of the graph clustering is linear in the number of edges.

**2. Convert to line graph.** The complexity of line graph conversion 2 is dominated by the nested loops for adding the edges to the line graph, resulting in a total complexity of $\mathcal{O}(|\mathcal{V}| * \deg_{avg}(\mathcal{V}))^2$, where $\deg_{avg}(\mathcal{V})$ is the average node degree. In the 1DSfM dataset, the average node degree is 45. The function is implemented in the NetworkX library (Hagberg et al., 2004–2024) and can be easily parallelized to convert each edge from the original graph to a node in the line graph.

**3. Create edge attributes.** Assigning relative rotations and relative translations to every edge in the line graph 3 scales linearly with the number of edges in the line graph, implying complexity $\mathcal{O}(|\mathcal{E}|)$.

**4. Create node attributes.** To create the node attribute 4, we stack the relative rotations and positions in the linegraph $L(\mathcal{G})$ into a tensor. The output is a node attribute tensor, created in $\mathcal{O}(|\mathcal{V}_L|) = \mathcal{O}(|\mathcal{E}|)$ time.

**5. Graph Neural Network Inference.** We run the GNN to classify all nodes in the line graph. Each layer computes the scaled dot-product attention for all nodes, where each node calculates the attention from all its neighbors. Therefore, the node update has a time complexity dominated by $\mathcal{O}(\deg_{avg}(\mathcal{V}_L) * |\mathcal{V}_L|) = \mathcal{O}(\deg_{avg}(\mathcal{V}_L) * |\mathcal{E}|)$, where $\deg_{avg}(\mathcal{V}_L)$ is the average node degree in the line graph.

Table 8: **Runtime** in minutes using the baseline based on (Manam & Govindu, 2022) abbreviated as MSAC score, CleanNet (Purkait et al., 2020), Triangle filter (Manam & Govindu, 2024), 1DSfM (Wilson & Snavely, 2014), 1DSfM combined with Triangle Filter and our proposed method, both standalone and in conjunction with the Triangle filter. Additionally, the results of the Oracle filter, removing all outlier translations, are also presented.

| Method | MSAC score | CleanNet | Triangle | 1DSfM | 1DSfM + Tri. | Ours | Ours+Tri. | Oracle |
|---|---|---|---|---|---|---|---|---|
| PhotoTourism dataset (minutes (↓)) | | | | | | | | |
| B. Museum | 0.03 | 0.06 | 0.37 | 0.02 | 0.36 | 4.87 | 5.19 | 0.21 |
| F. Cathedral | 0.00 | 0.06 | 0.00 | 0.00 | 0.00 | 0.09 | 0.09 | 0.00 |
| L. Memorial | 0.04 | 0.06 | 0.34 | 0.02 | 0.30 | 4.81 | 5.03 | 0.23 |
| M. Cathedral | 0.00 | 0.00 | 0.00 | 0.00 | 0.00 | 0.12 | 0.13 | 0.01 |
| M. Rushmore | 0.00 | 0.00 | 0.00 | 0.00 | 0.00 | 0.16 | 0.16 | 0.01 |
| Reichstag | 0.00 | 0.00 | 0.00 | 0.00 | 0.00 | 0.08 | 0.08 | 0.00 |
| Sacre Coeur | 0.13 | 0.10 | 0.65 | 0.03 | 0.59 | 8.68 | 8.96 | 0.37 |
| S. Familia | 0.01 | 0.02 | 0.04 | 0.01 | 0.04 | 1.01 | 1.04 | 0.06 |
| St. P. Cathedral | 0.03 | 0.05 | 0.26 | 0.01 | 0.25 | 3.33 | 3.45 | 0.17 |
| St P. Square | 0.51 | 0.37 | 11.45 | 0.16 | 10.93 | 62.90 | 64.35 | 1.48 |
| AVG | 0.08 | _0.07_ | 1.31 | **0.03** | 1.25 | 8.60 | 8.85 | 0.25 |
| 1DSfM dataset (minutes (↓)) | | | | | | | | |
| Alamo | 0.01 | 0.02 | 0.02 | 0.00 | 0.02 | 1.34 | 1.38 | 0.06 |
| E. Island | 0.00 | 0.01 | 0.00 | 0.00 | 0.00 | 1.29 | 1.31 | 0.01 |
| Gendarm. | 0.01 | 0.01 | 0.00 | 0.00 | 0.00 | 1.03 | 1.05 | 0.03 |
| M. Metro | 0.00 | 0.00 | 0.00 | 0.00 | 0.00 | 0.62 | 0.63 | 0.01 |
| M. N. Dame | 0.01 | 0.01 | 0.00 | 0.00 | 0.01 | – | – | 0.03 |
| NYC Library | 0.00 | 0.01 | 0.00 | 0.00 | 0.00 | 1.17 | 1.19 | 0.02 |
| N. Dame | 0.08 | 0.07 | 0.34 | 0.02 | 0.32 | 6.85 | 7.00 | 0.27 |
| P. del Popolo | 0.00 | 0.01 | 0.00 | 0.00 | 0.01 | 1.73 | 1.76 | 0.01 |
| Piccad. | 0.03 | 0.04 | 0.04 | 0.02 | 0.05 | 5.88 | 5.99 | 0.14 |
| R. Forum | 0.01 | 0.01 | 0.00 | 0.00 | 0.01 | 2.21 | 2.24 | 0.04 |
| T. of London | 0.00 | 0.01 | 0.00 | 0.00 | 0.00 | 1.17 | 1.19 | 0.02 |
| Trafalgar | 0.08 | 0.04 | 0.06 | 0.03 | 0.10 | 8.08 | 8.22 | 0.15 |
| U. Square | 0.02 | 0.01 | 0.00 | 0.00 | 0.00 | 1.10 | 1.11 | 0.02 |
| V. Cath. | 0.08 | 0.02 | 0.02 | 0.01 | 0.02 | 2.24 | 2.28 | 0.06 |
| Yorkminster | 0.00 | 0.00 | 0.00 | 0.00 | 0.00 | 1.62 | 0.80 | 0.01 |
| AVG | _0.02_ | _0.02_ | 0.03 | **0.01** | 0.04 | 2.59 | 2.64 | 0.07 |
| ScanNet dataset (minutes (↓)) | | | | | | | | |
| 0000 | 0.12 | 0.15 | 0.25 | 0.06 | 0.20 | 14.21 | 14.47 | 0.92 |
| 0059 | 0.02 | 0.03 | 0.02 | 0.01 | 0.02 | 2.77 | 2.82 | 0.18 |
| 0106 | 0.05 | 0.03 | 0.02 | 0.01 | 0.02 | 2.90 | 2.95 | 0.19 |
| 0169 | 0.04 | 0.04 | 0.02 | 0.01 | 0.02 | 3.76 | 3.83 | 0.20 |
| 0181 | 0.01 | 0.02 | 0.00 | 0.00 | 0.01 | 1.34 | 1.37 | 0.08 |
| 0207 | 0.01 | 0.01 | 0.00 | 0.00 | 0.01 | 1.56 | 1.59 | 0.08 |
| AVG | _0.04_ | 0.05 | 0.05 | **0.02** | _0.04_ | 4.42 | 4.50 | 0.28 |

Table 9: Running time in minutes for reconstruction using the **COLMAP** and **Theia** pipelines on 11 scenes of the 1DSfM dataset. The feature detection and matching times are not included in the runtimes.

| Scene | Alamo | E. Isl. | Gendar. | M. Metro | N. Dame | P. del Pop. | R. Forum | T. of London. | U. Square | V. Cath. | Yorkm. | AVG |
|---|---|---|---|---|---|---|---|---|---|---|---|---|
| Time in minutes (↓) | | | | | | | | | | | | |
| Theia | **27** | **12** | **21** | **9** | **61** | **13** | **32** | **13** | **12** | **34** | **14** | **23** |
| COLMAP | 2039 | 496 | 819 | 99 | 1800 | 306 | 561 | 592 | 596 | 1167 | 1272 | 886 |

---

**Algorithm 1** Graph Clustering

1: **Input:** view graph $\mathcal{G} = (\mathcal{V}, \mathcal{E})$, weights
2: $k \leftarrow 3$                    ▷ initial cluster number $\mathcal{O}(1)$
3: **while** $(\max(|\mathcal{E}_L|) >$ max edges that fit into memory) **do**
4:     Solve $k$-way graph partitioning            ▷ using METIS $\mathcal{O}(|E|)$
5:     Build $k$ subgraphs                    ▷ $\mathcal{O}(|E|)$
6:     Compute $\max(|\mathcal{E}_L|)$ for linegraphs of all subgraphs: $|\mathcal{E}_L| = \frac{1}{2} \sum_{v \in \mathcal{V}} \deg(v)^2 - m$ ▷ $\mathcal{O}(|E|)$
7:     $k \leftarrow k + 1$                    ▷ $\mathcal{O}(1)$
8: **end while**
9: **Output:** $k$ clusters

---

**Algorithm 2** Line Graph Construction

---

1: **Input:** graph $\mathcal{G} = (\mathcal{V}, \mathcal{E})$
2: **for all** $v \in \mathcal{V}$ **do**                                                                    ▷ $\mathcal{O}(|\mathcal{V}|)$
3:     **for all** adjacent edges to $v$ **do**                                              ▷ $\mathcal{O}(\deg(v))$
4:         $\mathcal{V}_L \leftarrow$ add edge                                                            ▷ $\mathcal{O}(1$
5:     **end for**
6:     **for all** $a \in \mathcal{V}_L$ **do**                                                         ▷ $\mathcal{O}(\deg(v))$
7:         **for all** $b \in \mathcal{V}_L \setminus \{a\}$ **do**                                 ▷ $\mathcal{O}(\deg(v))$
8:             $\mathcal{E}_L \leftarrow$ add edge $(a, b)$ if they share a common node      ▷ $\mathcal{O}(1)$
9:         **end for**
10:     **end for**
11: **end for**
12: **Output:** line graph $L(\mathcal{G}) = (\mathcal{V}_L, \mathcal{E}_L)$

---

**Algorithm 3** Create Edge Attributes

---

1: **Input:** line graph $L(\mathcal{G}) = (\mathcal{V}_L, \mathcal{E}_L)$
2: identify common nodes for all edge pairs                                           ▷ $\mathcal{O}(\mathcal{E}_L)$
3: retrieve rotations and features for common nodes                          ▷ $\mathcal{O}(\mathcal{E}_L)$
4: concatenate into one tensor per edge                                               ▷ $\mathcal{O}(\mathcal{E}_L)$
5: stack into final edge attribute tensor                                             ▷ $\mathcal{O}(\mathcal{E}_L)$
6: **Output:** edge attribute tensor

---

**Algorithm 4** Create Node Attributes

---

1: **Input:** graph $L(\mathcal{G}) = (\mathcal{V}_L, \mathcal{E}_L)$, relative poses
2: **for all** $v \in \mathcal{V}_L$ **do**                                                            ▷ $\mathcal{O}(|\mathcal{V}_L|)$
3:     stack rel. rotations and rel. positions to tensor
4: **end for**
5: **Output:** node attribute tensor

---

## E  ADDITIONAL ABLATION STUDIES

To demonstrate the importance of global rotations, relative rotations, and relative translations as network inputs, we trained the network without each of these components individually in *Ours w/o rotations*, *Ours w/o rel. rotations* and *Ours w/o rel. translations*. The results are presented in table 10. Our method, using all inputs, halves the mean position error compared to *Ours w/o rel. rotations* and reconstructs more cameras with higher accuracy than *Ours w/o rel. translations*. We framed the proposed method as a filtering technique for translation averaging under the assumption that rotation averaging is generally a less complicated problem. While incorporating global rotations into the filtering process yields the best results, we observe improvements even in the absence of rotations - *Ours w/o rotations* achieves good mean position error and recall, suggesting that the filter can be applied prior to rotation averaging. We would like to highlight that a poor-quality view graph negatively impacting the quality of the reconstruction is a general limitation of global SfM, *not specific to translation filtering*. As demonstrated in the experiments, the proposed method actually reduces the sensitivity of SfM to noisy pose graphs by removing outlier edges.

## F  ADDITIONAL DESCRIPTION OF CLUSTERING

Algorithm 1 outlines the clustering procedure to ensure that all graphs fit into memory. We begin by initializing the cluster number $k$ to 3 to obtain the initial cluster labels. Next, we compute the number of edges in the line graph of each cluster. If the edge count of a subgraph exceeds the maximum allowable number, we increment $k$ by one and restart the procedure. The maximum number of edges can be set automatically based on the current hardware, for example, by testing multiple values and selecting the one that results in subgraphs fitting into memory.

Table 10: Ablation study averaged over scenes Tower of London, Mount Rushmore, and Ellis Island. We report the mean position errors, the recall at 10, and the number of cameras.

| | Mean Position Error (↓) | Recall@10 (↑) | # of cameras (↑) |
|---|---|---|---|
| 1DSfM | 51.74 | 48.75 | **740** |
| w/o line graph | 43.82 | 51.19 | 582 |
| Ours w/o rel. rotations | 50.93 | 49.07 | 525 |
| Ours w/o rel. translations | 23.19 | 47.62 | 459 |
| Ours w/o rotations | 33.98 | 49.92 | 485 |
| Proposed | **22.09** | **51.39** | 548 |

## G  ADDITIONAL RECONSTRUCTIONS

Visualization as shown in Figures 5 and 6. We used the default settings of Theia (Sweeney), now utilizing translation re-estimation, which was observed to reconstruct a larger number of points. For Gendarmenmarkt, the gate appears with enhanced details in comparison to alternative approaches. In scene Madrid Metropolis, the highlighted area in Ours+Triangle represents the spacing between the architecture better. In scene Tower of London, both the standalone method and its combination with the Triangle Filter distinctly show two walls, highlighted in the green circle. Union Square shows minimal variance across methods, though the COLMAP reconstruction displays fewer points. In the Phototourism dataset in Fig. 6, scene Reichstag shows little variations among methods, all capturing details effectively. The Florence Cathedral Side COLMAP reconstruction shows the left wall as upright. However, only our method, both on its own and when used with the Triangle Filter, accurately captures the wall's orientation and aligns with the COLMAP reference.

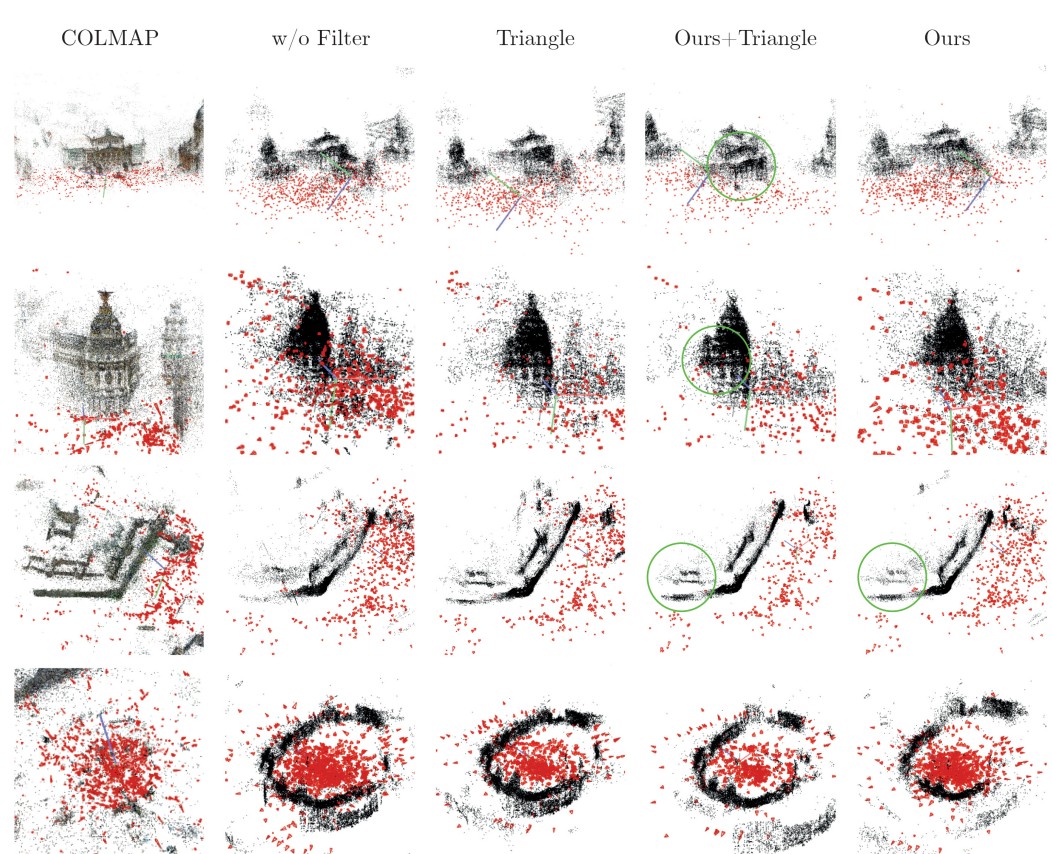

Figure 5: Reconstructions obtained by COLMAP  (Schonberger & Frahm, 2016) as reference and Theia's Global SfM pipeline (Sweeney), from left to right, utilizing no filter, our proposed filtering method, the combination of our method and Triangle Filter, and Triangle Filter standalone. The scenes from 1DSfM datasets top to bottom are Gendarmenmarkt, Madrid Metropolis, Tower of London and Union Square.

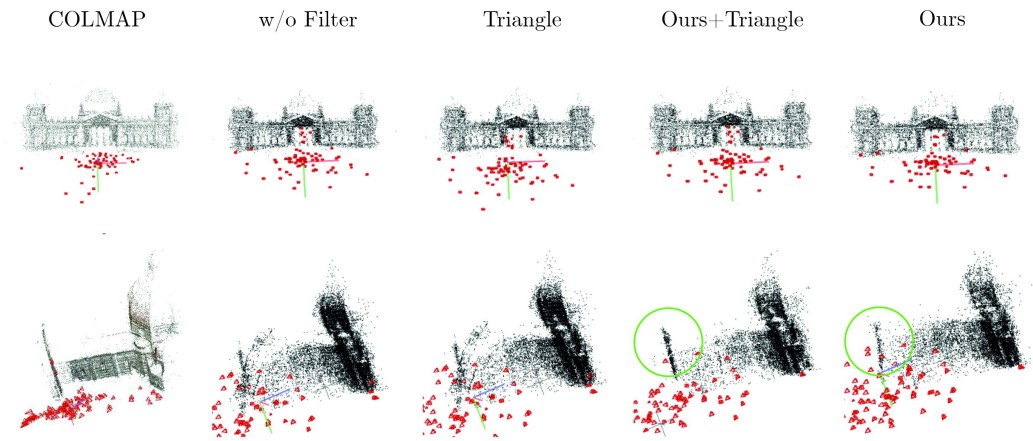

Figure 6: Reconstructions obtained by COLMAP  (Schonberger & Frahm, 2016) as reference and Theia's Global SfM pipeline (Sweeney), from left to right, utilizing no filter, our proposed filtering method, the combination of our method and Triangle Filter, and Triangle Filter standalone. The scenes from PhotoTourism datasets top to bottom are Reichstag and Florence Cathedral Side.

