# OpenReview forum: "Learning to Filter Outlier Edges in Global SfM"
_ICLR.cc/2025/Conference — ICLR 2025 Conference Withdrawn Submission_

### Official Review · Reviewer_bKhy · 2024-10-30

**Soundness:** 2
**Presentation:** 4
**Contribution:** 2
**Rating:** 3
**Confidence:** 4

**Summary:**

This paper introduces an innovative approach to improving camera position estimation in global Structure-from-Motion (SfM) frameworks. The method filters out erroneous edges—representing relative translations—in pose graphs before performing translation averaging. By framing this filtering process as a binary classification task within the dual graph, the authors leverage a Transformer-based model to effectively identify outlier edges. Their approach enhances camera position accuracy over existing filtering methods and integrates seamlessly with other filtering techniques.

**Strengths:**

1. Overall, the paper is well-written, providing essential definitions and insightful discussions.

2. It tackles a classic, important problem in the literature, applying a modern deep-learning approach.

3. The authors provide ablations of their design choices.

**Weaknesses:**

1. The method shows only slight accuracy improvements over other baselines, such as 1DSfM (Wilson & Snavely, 2014), particularly when considering its registration of fewer cameras and significantly higher runtime.

2. As this approach is framed as a classification task, it would benefit from including classification metrics to evaluate performance comprehensively. Including metrics like recall and precision for both inliers and outliers, especially on datasets such as 1DSfM, would provide a clearer assessment.

3. The paper introduces a filtering technique for translation averaging that can, in principle, be applied across different global SfM methods; however, it is tested solely within the Theia framework.

**Questions:**

1. It’s surprising that training is conducted on only a single scene. What makes you believe this is sufficient to achieve reliable results?

2. I would appreciate the inclusion of classification metrics, as detailed in the weaknesses section.

---

### Official Review · Reviewer_Mciz · 2024-11-01

**Soundness:** 2
**Presentation:** 3
**Contribution:** 2
**Rating:** 3
**Confidence:** 4

**Summary:**

For the translation averaging problem in global structure from motion, this paper proposes a learning-based method to filter out relative translation outliers in the pose graph and employs a graph clustering approach to prevent memory overflow. In experiment, the revised LUD algorithm is utilized to estimate camera positions from the filtered relative translations by different methods. The proposed method yields superior accuracy, to the standard outlier filtering techniques.

**Strengths:**

This paper introduces a learning-based approach to filter outlier edges for translation averaging. Specifically, given embeddings from DINO v2 and global rotations for each image, a Transformer architecture-based GCN and dual line graph are used to identify and filter outlier vertices.  Additionally, a graph clustering method is implemented to prevent memory overflow. Combining the proposed method with the Triangle filter improves camera position accuracy.

**Weaknesses:**

1. There is no available data on the angular error distribution of the filtered relative translations and relative translations after applying this method across different datasets. Such data would serve as a direct indicator of the filtering efficacy.

2. The median position error is more robust and convincing than the mean position error, as some optimized camera positions deviate significantly from the ground truth. Therefore, median position errors should be presented in the main paper—for instance, by including the median position error in Table 4.

3. For the task of translation averaging, both the number of estimated cameras and the accuracy of camera positions are crucial. As demonstrated in Table 6, there is a notable reduction in median errors when utilizing either the Triangle filter or the proposed method. However, this improvement also coincides with a significant reduction in the number of cameras. Relying on extensive filtering of cameras at the relative translation filtering stage to improve accuracy may not be advisable.

**Questions:**

1. For the raw pose graph as input, does this method also have a certain effect for other methods like BATA (Angle-based objective) or CReTA (Filter feature match outliers during optimization) ?

2. The state-of-the-art global structure-from-motion method GLOMAP asserts that incorporating relative translations in the position averaging step is unnecessary. Given this claim, why is the filtering of outlier edges considered a necessary step?

---

### Official Review · Reviewer_FuE4 · 2024-11-01

**Soundness:** 3
**Presentation:** 3
**Contribution:** 2
**Rating:** 5
**Confidence:** 4

**Summary:**

The paper proposes an outlier edge filter algorithm for the global SfM to improve the reconstruction robustness and accuracy. After the geometric verification and motion averaging, the scene graph is constructed where the edges represent the relative poses between images, where each node is extracted image features. Then, a graph neural network with Transformer is applied to classify where the edges are inliers or outliers. To reduce the computational cost, the authors propose to split the entire graph into several sub-graphs using a clustering approach. The proposed method is evaluated on multiple datasets including large-scale scenes and shows its efficacy in improving the reconstruction accuracy.

**Strengths:**

S1) The proposed graph outlier filter that leverages the geometrical relations between multiple images, as well as image features, is pretty novel.

S2) The overall writing is clear and easy to follow. The figure is well-organized and helpful to understand the proposed method.

S3) The proposed method is pretty efficient.

**Weaknesses:**

W1) Training. The proposed model is trained only on a single scene in the phototourism dataset. Even though the proposed method shows its efficacy on multiple test datasets, the reviewer thinks the method can be further trained on more diverse scenes for its generalization ability.

W2) Evaluation datasets. The reviewer thinks the proposed SfM methods can be evaluated on more commonly used SfM datasets with highly accurate ground truth camera poses, such as the ETH3D high-res dataset as used in Pixel-Perfect SfM.

W3) Performances. The reviewer finds that the proposed method shows a very slight improvement in the reconstruction accuracy compared to the baseline methods. For example, in Table 1 the best performance of mean positional error is 2.28 (paper) v.s. 2.35 (baseline), in Table 2 the best performance of mean positional error is 21.15 (paper) v.s. 22.61 (baseline), in Table 3 the best performance of mean positional error is 0.62 (paper) v.s. 0.66 (baseline). The reviewer thinks these results are not significant enough to show the superiority of the proposed method. The lack of training on diverse scenes may be a reason.

**Questions:**

Q1) The reviewer is curious about whether training the proposed model on more diverse scenes can further improve the reconstruction accuracy.

Q2) The repetitive structures in the scene graph may cause severe erroneous matches, as suggested in Doppelgangers[1]. The reviewer is curious about how the proposed graph filtering method handles this issue. Including more results on this type of dataset[2] may be helpful to show the robustness of the proposed method.

[1] Cai, Ruojin et al. “Doppelgangers: Learning to Disambiguate Images of Similar Structures.” 2023 IEEE/CVF International Conference on Computer Vision (ICCV) (2023): 34-44.

[2] Roberts, Richard et al. “Structure from motion for scenes with large duplicate structures.” CVPR 2011 (2011): 3137-3144.

---

### Official Review · Reviewer_gJvM · 2024-11-04

**Soundness:** 1
**Presentation:** 3
**Contribution:** 1
**Rating:** 3
**Confidence:** 4

**Summary:**

The paper works on edge filtering in Global SFM. The authors propose using a three-layer transformer to classify the dual graph, where the vertices are relative transformations and the edges are cameras. The global orientation and image embedding from DINO features are used as the feature for the graphical neural network with three TransformerConv layers. The proposed method was trained on a scene with 249 images. The authors compared the proposed method with multiple baselines on the PhotoTourism dataset, 1DSfM dataset, and ScanNet Dataset and achieved overall good results. The authors also provided visualizations on differences between different filtering method.

**Strengths:**

1. The paper is clearly written with a clear definition of the problem and technical contribution.
2. The proposed method takes advantage of DINO features, which avoids retraining the image perception model.
3. The model trained with limited data generalizes well across different datasets and achieves comparable results with previous methods.

**Weaknesses:**

1. The pose graph edge filtering problem is a submodule of the SfM system. The technical contribution is limited compared to proposing comprehensive solutions for the SfM system. Although the authors suggested the proposed method is applicable to other SfM systems, there are no experimental results to support it.
2. The improvements are marginal, and sometimes, the method performed worse than 1DSfM, although the proposed method takes much longer running time.
3. A model with three transformer layers and hundreds of training data does not seem technically sound. It possibly hints this is a simple method that typical machine learning methods could address very well (as suggested by baselines in the experiments).
4. As a follow-up, using the 3D orientation directly as the input of the neural network doesn't seem to be technically sound, given limited training data, as the model might be dependent on the size of the scene if it's not trained with equivalent networks.
5. There have been significant advancements in image matching with foundational models like Dust3R and its follow-ups. The use of DINO features is also questionable as it only uses global features of images.

**Questions:**

1. What is the format for the 3D global orientation $R_v$ when it's fed into the transformer?
2. What's the consideration of not using per-pixel features or dense matching results as the features for the classification problem?\
3. Is the model only trained on a scene with a few hundred images? Would a three-layer transformer with hundreds of dimensions of features be overkill for this data capacity?
4. Given the limited training data distribution, what if the proposed method is used for indoor SfM or other scenes with significant scale differences?
5. The results on M. Rushmore are the same across all methods in Table 1. Is this because edge filtering is not important in that scene?

---

### Note · Authors · 2024-11-13

I have read and agree with the venue's withdrawal policy on behalf of myself and my co-authors.